

# The IsoGenie database: an interdisciplinary data management solution for ecosystems biology and environmental research

Benjamin Bolduc[1,*], Suzanne B. Hodgkins[1,*], Ruth K. Varner[2,3], Patrick M. Crill[4], Carmody K. McCalley[5], Jeffrey P. Chanton[6], Gene W. Tyson[7], William J. Riley[8], Michael Palace[2,3], Melissa B. Duhaime[9], Moira A. Hough[10], IsoGenie Project Coordinators, IsoGenie Project Team, A2A Project Team, Scott R. Saleska[10], Matthew B. Sullivan[1,11] and Virginia I. Rich[1]

[1] Department of Microbiology, The Ohio State University, Columbus, OH, USA
[2] Earth Systems Research Center, Institute for the Study of Earth, Oceans and Space, University of New Hampshire, Durham, NH, USA
[3] Department of Earth Sciences, College of Engineering and Physical Sciences, University of New Hampshire, Durham, NH, USA
[4] Department of Geological Sciences and Bolin Centre for Climate Research, Stockholm University, Stockholm, Sweden
[5] Thomas H. Gosnell School of Life Sciences, Rochester Institute of Technology, Rochester, NY, USA
[6] Department of Earth, Ocean, and Atmospheric Science, Florida State University, Tallahassee, FL, USA
[7] Australian Centre for Ecogenomics, School of Chemistry and Molecular Biosciences, University of Queensland, Brisbane, QLD, Australia
[8] Climate and Ecosystem Sciences Division, Lawrence Berkeley National Laboratory, Berkeley, CA, USA
[9] Department of Ecology and Evolutionary Biology, University of Michigan, Ann Arbor, MI, USA
[10] Department of Ecology and Evolutionary Biology, University of Arizona, Tucson, AZ, USA
[11] Department of Civil, Environmental and Geodetic Engineering, The Ohio State University, Columbus, OH, USA
* These authors contributed equally to this work.

Corresponding authors
Benjamin Bolduc,
bolduc.10@osu.edu
Virginia I. Rich, rich.270@osu.edu

## ABSTRACT

Modern microbial and ecosystem sciences require diverse interdisciplinary teams that are often challenged in "speaking" to one another due to different languages and data product types. Here we introduce the IsoGenie Database (IsoGenieDB; https://isogenie-db.asc.ohio-state.edu/), a de novo developed data management and exploration platform, as a solution to this challenge of accurately representing and integrating heterogenous environmental and microbial data across ecosystem scales. The IsoGenieDB is a public and private data infrastructure designed to store and query data generated by the IsoGenie Project, a ~10 year DOE-funded project focused on discovering ecosystem climate feedbacks in a thawing permafrost landscape. The IsoGenieDB provides (i) a platform for IsoGenie Project members to explore the project's interdisciplinary datasets across scales through the inherent relationships among data entities, (ii) a framework to consolidate and harmonize the datasets needed by the team's modelers, and (iii) a public venue that leverages the same spatially explicit, disciplinarily integrated data structure to share published

datasets. The IsoGenieDB is also being expanded to cover the NASA-funded Archaea to Atmosphere (A2A) project, which scales the findings of IsoGenie to a broader suite of Arctic peatlands, via the umbrella A2A Database (A2A-DB). The IsoGenieDB's expandability and flexible architecture allow it to serve as an example ecosystems database.

## INTRODUCTION

Understanding and predicting the behavior of complex natural systems requires collecting and integrating data across multiple disciplines (*Michener & Jones, 2012*). While data from a single discipline offer a limited view into the system as a whole, interdisciplinary data integration can provide independent and complementary views of the same phenomenon, as well as illuminate emergent non-additive behaviors (*McCalley et al., 2014*; *Deng et al., 2017*; *Woodcroft et al., 2018*; *Wilson et al., 2019*). Interdisciplinary, systems-scale integration of data—from isotopic and other geochemical measurements, to measures of microbial ecology and biochemistry, to climate data, to vegetation surveys and remote sensing—is essential to modeling and predicting biogeochemical cycling (*Heffernan et al., 2014*; *Fei, Guo & Potter, 2016*; *Rose et al., 2017*).

First, some definitions are required before heterogenous data integration can be clearly discussed. For the purpose of this paper, *data* is broadly defined as information (usually numeric) collected using the scientific method, and a *dataset* is a structured collection of related data submitted by a person or lab as a cohesive entity. Examples of datasets include organic matter mass spectra with both compound-resolved and broad compound class relative abundances; a collection of DNA sequence raw reads, processed reads, and derived metagenome-assembled genomes (MAGs); and a timeseries of temperatures measured by a particular instrument at several locations. Datasets are not monolithic, but can be split or combined to address specific research questions; therefore, we use the term *data product* to refer to any structured collection of related data, whether a dataset, a piece of a dataset, or a collection of related datasets. Similarly structured data products from the same discipline (e.g., two different DNA sequence datasets from different labs) are referred to as being of the same *data product type* (or, for datasets specifically, *dataset type*). Data *format* describes the organization of a data product, and encompasses both file format (e.g., .csv, .fastq) and the data organization within files of a given format (e.g., the specific columns present within a spreadsheet). *Data processing* is any kind of quality control or calculation performed upon data, including the conversion of raw data to ecologically-relevant "derived" products. This includes processes such as simple blank correction, calibration of peak heights to determine chemical concentrations, quality control of sequencing reads, and creation of vegetation maps from remote sensing images. Processing may result in increased *data quality*, defined as the degree to which a data

product represents an unbiased yet intuitively understandable snapshot of reality. *Data integration* is a specific type of processing in which different datasets—often across disciplines—are combined to enable exploration and synthesis. An organized framework containing data is a *data structure*, and its organizational paradigm is a *data model*. An integrative data structure usually requires creating small subsets of data based on location, timeframe, or other criteria, to allow recombination (integration); we refer to these subsets as *data entities*, and these can be conceptualized as sets of key: value pairs (e.g., a set of concentrations of chemical species for one sample). Finally, *metadata* describes information necessary to understand a data entity's meaning and origins. For our purposes, metadata falls under two categories: *File metadata* includes information about the source file from which a data entity derives, including the file path, contact information of the investigator(s), and variable definitions and units. *Sampling metadata* includes information about the physical origin of a data entity, such as the name of the site, GPS location, sampling date, and depth. Unlike file metadata, which is often stored in a "readme" file for a dataset, sampling metadata is typically stored within individual table rows under "descriptor" columns.

Organizing and connecting heterogenous data is not a new challenge, and a number of online repositories have been designed for this purpose, such as ESS-DIVE (https://ess-dive.lbl.gov/), NEON (https://www.neonscience.org/), LTER (https://lternet.edu/), PANGAEA (https://www.pangaea.de/), DataDryad (https://datadryad.org/), and DataONE (https://www.dataone.org/), among others. These systems are powerful and have their respective strengths, such as allowing datasets to be filtered based on keywords, locations, and other metadata. However, they do not *integrate their data in a framework that allows scalable and detailed querying* (e.g., quickly extracting all water table and temperature data from multiple sites into a single table, for scaling of water table-temperature relationships from individual sites to a broader geographical range). The field of bioinformatics is further along in this regard: for molecular meta-omic data, numerous databases (e.g., MIGS/MIMS, MIMAS, IMG/M, GeneLab) (*Hermida et al., 2006*; *Field et al., 2008*; *Gattiker et al., 2009*; *Chen et al., 2019*; *Ray et al., 2019*) and integrative data management platforms (e.g., KBase, MOD-CO, ODG, GeNNet, BioKNO, MGV, OMMS, mixOmics) (*Sujansky, 2001*; *Symons & Nieselt, 2011*; *Perez-Arriaga et al., 2015*; *Yoon, Kim & Kim, 2017*; *Costa et al., 2017*; *Rohart et al., 2017*; *Guhlin et al., 2017*; *Manzoni et al., 2018*; *Arkin et al., 2018*; *Brandizi et al., 2018*; *Rambold et al., 2019*) have been developed, and often include standardization of sample metadata to enable efficient data integration. Notable among these are KBase (https://kbase.us/) (*Arkin et al., 2018*), which provides "apps" through which users can process their data in a framework that tracks processing steps ("provenance") in an accessible format, and MOD-CO (*Rambold et al., 2019*), a bioinformatics data processing tool that includes a conceptual schema and data model to track metadata and workflows. However, these tools are built around molecular-scale biological data and do not attempt to describe the highly diverse, multi-scale heterogenous data typical of landscape-scale ecological studies. A few projects, to date, have attempted this: One of the most comprehensive of these is the ISA framework (https://www.isacommons.org/)

(*Sansone et al., 2012*), a set of interrelated data annotation and analysis tools designed for life and environmental science. This framework is based around the ISA Abstract Model, a formalized metadata storage model that links samples, processes, and datasets using directed acyclic graphs stored in tabular or JSON format. However, the storage of graphs in separate table-like files precludes the fast querying capabilities of graph databases, which are more typically used for storing this type of data structure. Similarly, *Barkstrom (2010)* developed a system to track metadata relating to Earth science datasets' authorship and processing histories, but this system focuses mainly on metadata, rather than ability to query and explore the data themselves. The IndexMed Consortium (*David et al., 2015*) was formed to describe ecological data from the Mediterranean and attempted to integrate heterogenous data by linking multiple databases in a decentralized indexed framework. However, its decentralized model poses challenges for standardizing heterogenous data (which is needed for efficient cross-comparison), and its website no longer exists, nor is its codebase readily available to other teams. Several biodiversity data platforms (reviewed in *Triebel, Hagedorn & Rambold (2012)*) have explored data sharing and cross-linkages, but the platforms used to store different dataset types (e.g., species distribution and sequencing data) remain separate.

*Integrating* data—and doing so across disciplines, beyond simple cataloging and indexing—becomes critically important as data analysis methods become increasingly sophisticated (*Michener & Jones, 2012*). As technological innovation allows, research is increasingly being performed across broader scales (e.g., from satellite data to processes at the micro and nanoscale) (*Peters et al., 2008*; *Heffernan et al., 2014*; *Fei, Guo & Potter, 2016*; *Rose et al., 2017*; *Jansen et al., 2019a*), and new and more diverse data are being generated. This has prompted international efforts to develop standards that make data more Findable, Accessible, Interoperable, and Reusable among different labs (FAIR principles) (*Wilkinson et al., 2016*; *Brandizi et al., 2018*). These include programs, specifications, and ontologies such as Ecological Metadata Language (EML) (https://eml.ecoinformatics.org/) (*Jones et al., 2019*), ABCD (https://abcd.tdwg.org/) (*Access to Biological Collection Data task group, 2007*), INSPIRE (https://inspire.ec. europa.eu/), and Environmental Ontologies (ENVO) (http://www.obofoundry.org/ ontology/envo.html) (*Buttigieg et al., 2013*, *2016*) developed for environmental data; the MIGS/MIMS, MIMARKS, and MIxS specifications (*Field et al., 2008*; *Yilmaz et al., 2011*) developed by the Genomic Standards Consortium (https://press3.mcs.anl.gov/gensc/) for biological sequencing data; and ISO standards (https://www.iso.org/standards.html) developed to standardize procedures for a broad range of applications including environmental management. Integrating and organizing highly diverse data, particularly when the data are derived across a broad set of disciplines, presents numerous technical and conceptual challenges.

Challenges to data integration primarily relate to data processing complexities. *First*, data from different disciplines often exist in different formats that do not compare directly, or the same measurement (e.g., gas flux) lacks standardization of units or sampling resolution across labs. *Second*, researchers start from raw measurements, but there are

often numerous derived data products at different stages of quality control and processing, and these vary across disciplines. These derived data products may also be housed in distinct accepted data organization platforms; for example, for biomolecule sequence data, the Sequence Read Archive (SRA) (*Leinonen, Sugawara & Shumway, 2011*) houses raw genomic and transcriptomic sequence reads, GenBank (*Benson et al., 2012*) houses MAGs and other derived data, and separate repositories exist for proteome data, including PRIDE (*Martens et al., 2005*; *Vizcaíno et al., 2016*) and ProteomeXchange (*Vizcaíno et al., 2014*). *Third*, beyond process-related quality control (to remove poor data), there is variation in data "quality levels" resulting from original experimental design and/or assumptions made during processing, and these vary across disciplines. For example, a chemical spectral dataset may first undergo blank and background corrections to improve quality level, and then be used to generate summary metrics that capture more ecologically relevant chemical properties, but these summary metrics may involve assumptions that compound empirical uncertainty (*Ishii & Boyer, 2012*). Full understanding of these discipline-specific experimental quality standards is critical to integrating and comparing data across disciplines. *Fourth*, as technologies improve, data quality changes, which for long-term projects necessitates raw data retention to enable systematic reprocessing for modernized synthesis products. This can result in numerous versions of processed data, each with a different quality level, for the same set of raw measurements.

Given these challenges, an interdisciplinary data integration platform must include a standardized system for versioning datasets, labeling them by quality, and tracking data collection and processing workflows. While the first two tasks can be easily accomplished with standardized version numbers and quality labels, the high variety of data collection and processing workflows across disciplines necessitates a structure that can accommodate this variety without relying entirely on textual methods descriptions. The most straightforward way to accomplish this is *for the data structure to mimic the physical relationships between physical entities*: specifically, the relationships connecting physical entities (such as sampling sites, soil cores collected from and instruments located at these sites, and the resulting connections between data from these related entities) should be recapitulated in a data structure that maps these physical entities (and their connecting relationships) to corresponding data entities. Such a structure is not only intuitively understandable across disciplines, it also advances data organization in ways that facilitate exploration of ecological, organismal, and physicochemical interactions occurring both horizontally and vertically. This presents its own conceptual challenges, the most fundamental of which is *how* data relate within a data model or structure. In other words, while apples cannot strictly be compared to pears, traits they could plausibly share include collection location, assay technique, or properties of a higher-level classification; for example, these fruit types can be represented as properties of "fruit" objects.

Here we sought to leverage diverse datasets generated by a large-scale international ecosystem science study, IsoGenie, to establish an integrative, interdisciplinary graph database platform, the IsoGenie Database (IsoGenieDB). This platform brings together

ideas from existing frameworks (e.g., metadata provenance tracing: *Barkstrom (2010)*; ISA framework: *Sansone et al. (2012)*; IndexMed Consortium: *David et al. (2015)*; KBase: *Arkin et al. (2018)*; MOD-CO: *Rambold et al. (2019)*) by integrating data from multiple disciplines in a way that mirrors the physical relationships and workflow between data entities, while also allowing detailed exploration and fast querying of the data themselves.

## MATERIALS AND METHODS

### Assembly of interdisciplinary project datasets in need of integration

The IsoGenie Project (Fig. 1) is a DOE-funded, interdisciplinary international team studying ecosystem climate feedbacks of Stordalen Mire (68°22′ N, 19°03′ E; Fig. 2A), a thawing permafrost system located near Abisko, in northern Sweden. This long-term study site offers historical data spanning more than a century, providing the foundation for IsoGenie's systems characterization using modern technical approaches. Data from the IsoGenie and related projects (Table 1; Fig. 1A) span multiple levels of study, with temporal scales ranging from minute- to decadal-resolution, and spatial scales ranging from nanoscale (e.g., elemental composition of soil and pore water), to microscale (e.g., microbial composition and metabolic processes), to macroscale (e.g., vegetation surveys and drone and satellite imagery) (Fig. 1B). By integrating these myriad types of datasets, the project has characterized how thaw-induced changes in hydrology and vegetation (*Malmer et al., 2005*; *Johansson et al., 2006*; *Bäckstrand et al., 2010*; *Palace et al., 2018*) drive changes in organic matter (*Hodgkins et al., 2014*, *2016*; *Wilson et al., 2017*; *Wilson & Tfaily, 2018*) and microbial and viral communities (*Mondav et al., 2014*; *Trubl et al., 2016*, *2018*, *2019*; *Singleton et al., 2018*; *Emerson et al., 2018*; *Woodcroft et al., 2018*; *Martinez et al., 2019*; *Wilson et al., 2019*; *Roux et al., 2019*), giving rise to changes in carbon gas emissions (*Wik et al., 2013*, *2018*; *Hodgkins et al., 2014*, *2015*; *McCalley et al., 2014*; *Burke et al., 2019*; *Perryman et al., 2020*), and collectively these insights are allowing improvements in predictive models (*Deng et al., 2014*, *2017*; *Chang et al., 2019a*, *2019b*; *Wilson et al., 2019*).

The IsoGenie Project interfaces with several related projects at Stordalen Mire via inter-project sharing of public datasets. These other projects include an NSF-funded Northern Ecosystems Research for Undergraduates (REU) program, which focuses on mapping changing vegetation (*Palace et al., 2018*), lake sediment control on $CH_4$ emissions from post-glacial lakes (*Wik et al., 2018*), and emissions from thaw ponds (*Burke et al., 2019*) in the Mire. Another study, on changes in organic matter decomposition along several permafrost thaw transects (*Malhotra & Roulet, 2015*; *Malhotra et al., 2018*), includes sites that have recently been incorporated into the extensive set of Stordalen sites sampled by IsoGenie. Finally, Stordalen Mire is one of five peatlands investigated by the NASA-funded Archaea to the Atmosphere (A2A) Project, whose goal is to upscale the IsoGenie Project's findings relating microbial communities and $CH_4$ emissions to the pan-Arctic.

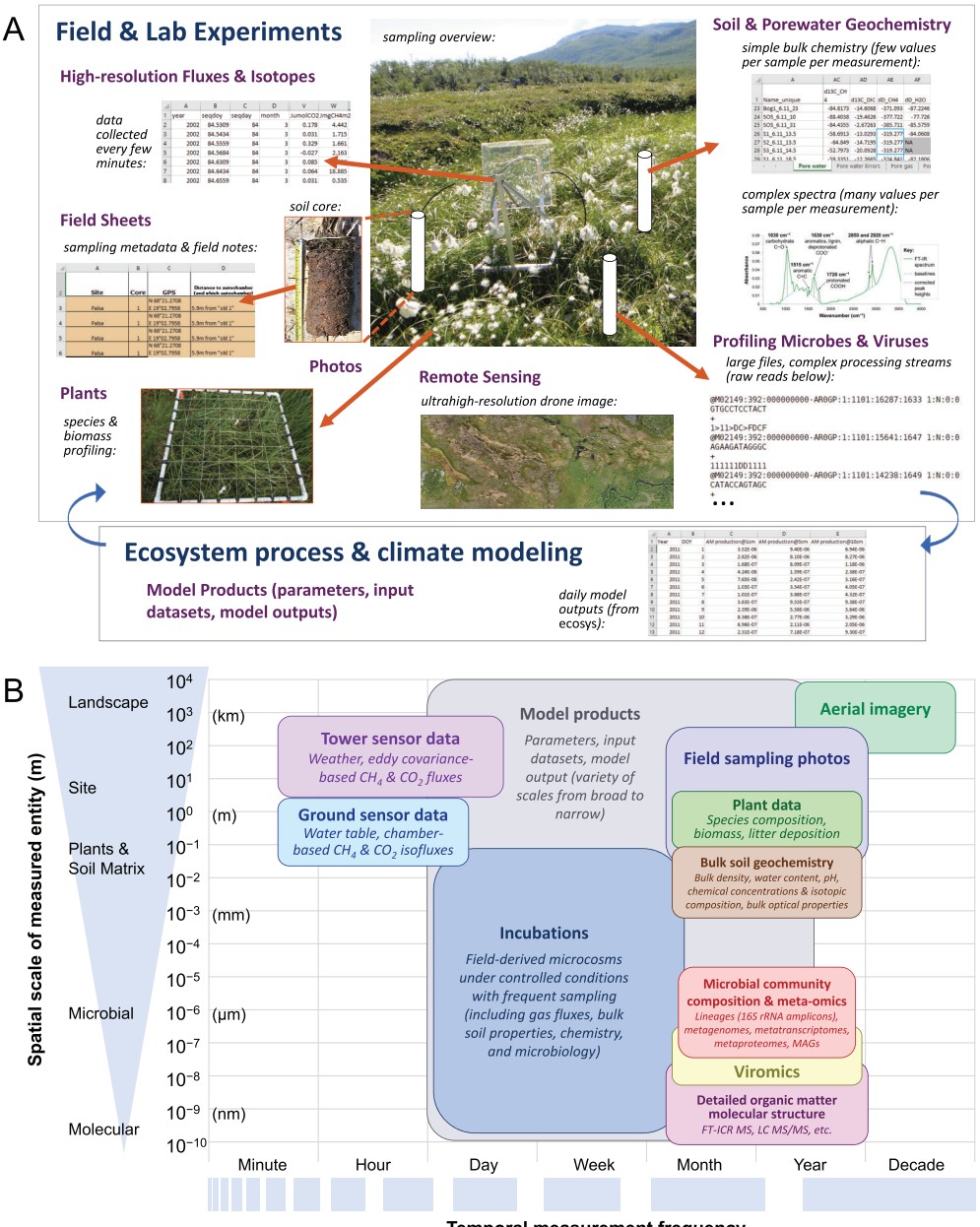

**Figure 1** The IsoGenie Project's systems approach to characterizing carbon cycling, involving integration of diverse data spanning a broad range of spatial and temporal scales. (A) General overview of project approach and dataset types arising (purple text), with screenshots, figures, or other images shown for select dataset types. Image sources available on the IsoGenieDB website (https://isogenie-db.asc.ohio-state.edu/): The FTIR spectrum figure (under "complex spectra…") is from *Hodgkins (2016)*. All spreadsheet and text file screenshots are of files available on the Downloads page under similar headings. Photo credits: "sampling overview," Nicole Raab; "soil core," Virginia Rich; "species & biomass profiling," Moira Hough; "ultrahigh resolution drone image," Michael Palace (*Palace et al., 2018*). (B) A more comprehensive list of dataset types oriented along spatial and temporal scale axes. OTUs, operational taxonomic units; MAGs, metagenome assembled genomes; FT-ICR MS, Fourier transform ion cyclotron resonance mass spectrometry; LC MS/MS, liquid chromatography with tandem mass spectrometry.
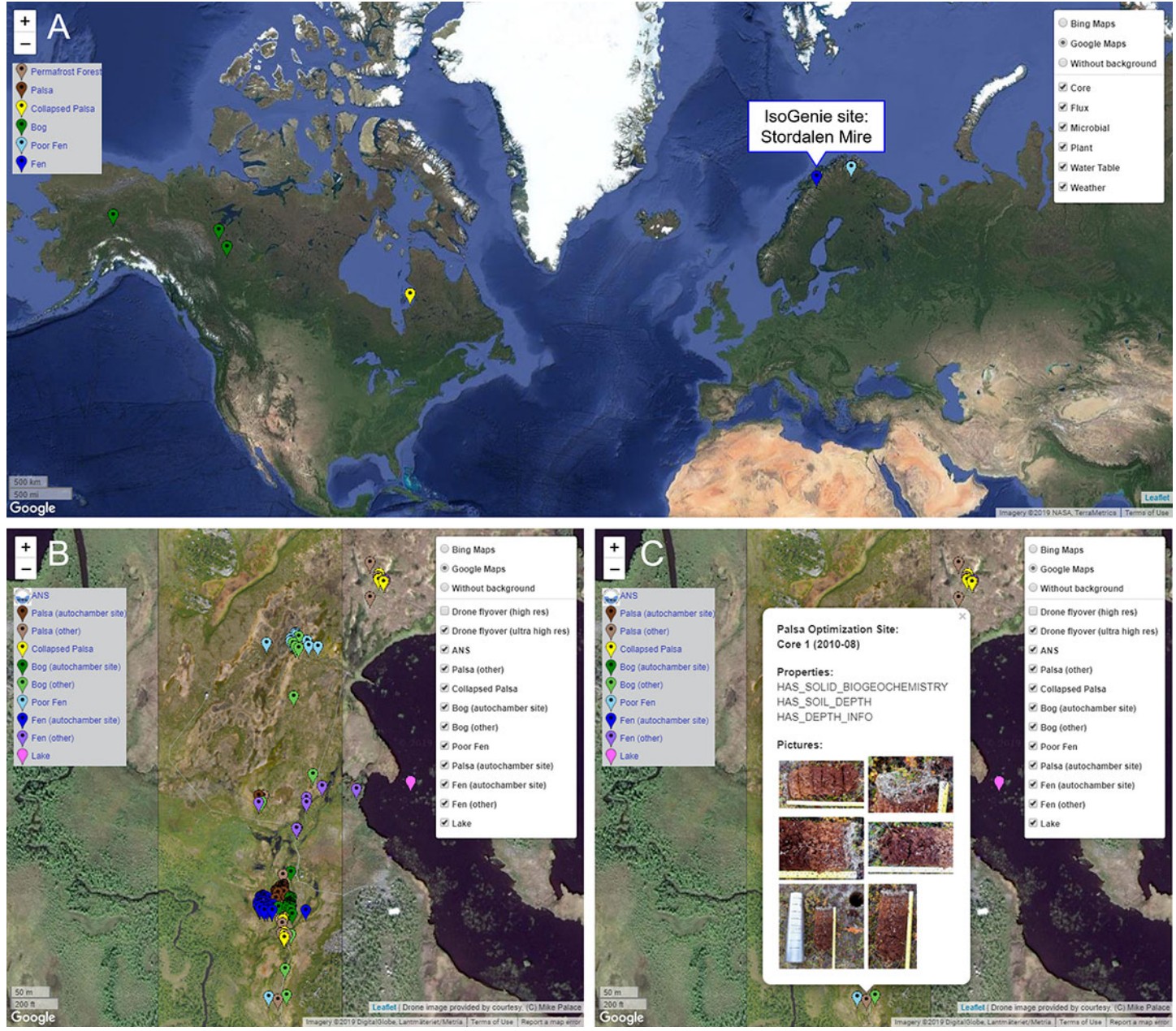

**Figure 2** Browsing sites through the Map Interface on the IsoGenieDB website (https://isogenie-db.asc.ohio-state.edu/maps). (A) Location of the IsoGenie Project site (Stordalen Mire), shown on the A2A-DB version of the Map Interface (https://a2a-db.asc.ohio-state.edu/maps) with four other comparator Arctic peatland sites. (B) The IsoGenieDB Map Interface shows the core locations (color-coded by habitat type) overlain on an ultra-high-resolution drone image (*Palace et al., 2018*). (C) Each point includes a popup with the site and core name, sampling date, available data, and core images. Background maps: (A) © 2019 Google, NASA, TerraMetrics; (B and C) © 2019 Google, DigitalGlobe, Lantmäteriet/Metria.

## Building a database to integrate across the heterogenous data generated by IsoGenie

The diversity of disciplines, dataset types, and spatial and temporal scales studied by IsoGenie (Fig. 1) presents a unique data integration challenge. We sought to address this
**Table 1 Summary of dataset types in the IsoGenieDB.**

| | Public? | References |
|---|---|---|
| *Sampling information:* | | |
| Coring metadata (sampling dates, locations, depths, samples taken) | X | Numerous |
| Field data (water table and active layer depths, pH, and air and soil temperatures at time of coring) | X | Numerous |
| *Weather and other environmental data:* | | |
| ANS daily and hourly weather summaries (including daily weather from 1913–present) | | *Callaghan et al. (2010)*, *Jonasson, Johansson & Christensen (2012)*, *Jonasson et al. (2012)*, and numerous others |
| Mire weather, soil temperature and moisture profiles, heat fluxes, and NDVI (continuous) | | *Jansen et al. (2019a, 2019b)* and *Garnello (2017)* |
| Lake water temperature profiles (continuous) | X | *Wik et al. (2013)* |
| *Gas fluxes:* | | |
| Total hydrocarbon, $CH_4$, and $CO_2$ fluxes from autochambers | X | *Bäckstrand et al. (2008a, 2008b, 2010)*, *Jackowicz-Korczyński et al. (2010)* and *Holst et al. (2010)* |
| $\delta^{13}C$ values of $CH_4$ and $CO_2$ fluxes from autochambers | | *McCalley et al. (2014)* |
| Thaw pond bubble fluxes and associated temperatures | X | *Burke et al. (2019)* |
| *Terrestrial subsurface geochemistry:* | | |
| $CH_4$ and $CO_2$ concentrations and $\delta^{13}C$ values | X | *McCalley et al. (2014)*, *Hodgkins et al. (2015)*, *Hodgkins (2016)* and *Perryman et al. (2020)* |
| Dissolved species concentrations (DOC, TN, acetate and other VFAs, $O_2$, $PO_4^{3-}$, $SO_4^{2-}$, $NO_3^-$, $NH_3$, Mn, Fe, Ca, Mg) | X | *Hodgkins (2016)* and *Perryman et al. (2020)* |
| Peat water content | X | *Hodgkins (2016)* |
| Peat bulk density | | |
| Peat C and N concentrations, $\delta^{13}C$ and $\delta^{15}N$, and C/N ratios | X | *Hodgkins et al. (2014)* and *Hodgkins (2016)* |
| Radiocarbon ages of peat and DIC | X | *Hodgkins (2016)* |
| FT-ICR MS | X (summarized indices) | *Tfaily et al. (2012)*, *Hodgkins et al. (2014, 2016)*, *Hodgkins (2016)*, *Wilson et al. (2017)* and *Wilson & Tfaily (2018)* |
| DOM optical properties (UV/Vis and EEMS) | X (summarized indices) | *Hodgkins (2016)* and *Hodgkins et al. (2016)* |
| Peat FTIR | X (summarized indices) | *Hodgkins et al. (2014, 2018)* |
| Results from peat incubations ($CH_4$ and $CO_2$ production and $\delta^{13}C$ values) | X (summarized indices) | *Hodgkins et al. (2014, 2015)*, *Hodgkins (2016)*, *Wilson et al. (2017, 2019)*, and *Perryman et al. (2020)* |
| *Microbial and viral sequencing:* | | |
| 16S rRNA amplicons (~100) | X | *McCalley et al. (2014)*, *Deng et al. (2017)*, *Mondav et al. (2017)*, *Martinez et al. (2019)* and *Wilson et al. (2019)* |
| Metagenomes (~375) | X | *Mondav et al. (2014)*, *Singleton et al. (2018)*, *Woodcroft et al. (2018)* and *Martinez et al. (2019)* |
| Metatransciptomes (24) | X | *Singleton et al. (2018)*, *Woodcroft et al. (2018)* and *Martinez et al. (2019)* |
| Metaproteomes (16) | X | *Mondav et al. (2014)*, *Woodcroft et al. (2018)* and *Martinez et al. (2019)* |
| Viromes (65) | X | *Trubl et al. (2016, 2018, 2019)* and *Roux et al. (2019)* |
| Viral population contigs mined from metagenomes | X | *Emerson et al. (2018)* |

(Continued)

| | Public? | References |
|---|---|---|
| Table 1 (continued) | | |
| MAGs (~1600) | X | *Singleton et al. (2018)*, *Woodcroft et al. (2018)* and *Martinez et al. (2019)* |
| Lake sediment microbial community composition (based on 16S rRNA amplicons, metagenomes, and MAGs) | X | J.B. Emerson et al., 2020, unpublished data |
| *Plant-associated data:* | | |
| Ultra-high-resolution drone imagery | X (as overlay on Map Interface) | *Palace et al. (2018)* |
| Rhizosphere and phyllosphere microbial community composition (iTag-based) | X | M.A. Hough et al., 2019, unpublished data |
| *Modeling:* | | |
| *Ecosys* model input data, code, and outputs | X | *Chang et al. (2019a, 2019b)* |

**Notes:**
Listed dataset types include citations of IsoGenie and related project papers that use each type of dataset, and those marked with an "X" are available in the public portion of the DB. ANS, Abisko Naturvetenskapliga Station; NDVI, normalized difference vegetation index; DOC, dissolved organic carbon; TN, total nitrogen; VFAs, volatile fatty acids; DIC, dissolved inorganic carbon; FT-ICR MS, Fourier transform ion cyclotron resonance mass spectrometry; DOM, dissolved organic matter; UV/Vis, ultraviolet/visible absorption spectroscopy, EEMS, excitation-emission matrix spectroscopy; FTIR, Fourier transform infrared spectroscopy; MAGs, metagenome-assembled genomes.

challenge by developing the IsoGenie Database (IsoGenieDB), a novel data management and data exploration platform that integrates these heterogenous data into the same data structure. The IsoGenieDB is powered by Neo4j, a graph database platform that is also used by NASA, Walmart, eBay, Adobe, and numerous other well-established organizations, helping ensure its reliable longevity. The IsoGenieDB uses data entities and the inherent relationships between them as the basic building blocks of the database. These building blocks are used independent of data source, connecting all the data within one integrated structure.

The IsoGenieDB is built with a set of custom tools, available at https://bitbucket.org/MAVERICLab/isogeniedb-tools/, that use a combination of the py2neo Python package (https://py2neo.org) and Neo4j's own "Bolt" Python driver (https://neo4j.com). The fundamental design of the database follows a property graph model (*Miller, 2013*). Data entities are stored within *nodes*, which serve as the primary unit of organization. Nodes can have *labels*, which serve as a high-level means of categorizing nodes for fast access and classification (e.g., a node representing a bog soil core has the labels *Bog* and *Core*; hereafter, specific labels are referred to with capitalized words). Data and other detailed information are stored within the nodes as a set of key: value *properties* (e.g., *Site__:Bog1* and *Date__:2013-07-14*). These properties can contain any numeric or text-based information (e.g., time, temperature, pH), as well as links to flat files that store non-text information (e.g., core photographs, raw instrument trace outputs). Nodes are connected to other nodes through *relationships* (also known as edges), which store information about how the two nodes are related.

The IsoGenieDB leverages the inherent relationships within the data (e.g., commonalities in sampling location or measurement technique) to build the basic structure of the database. In the IsoGenieDB, most of the nodes are organized hierarchically based on

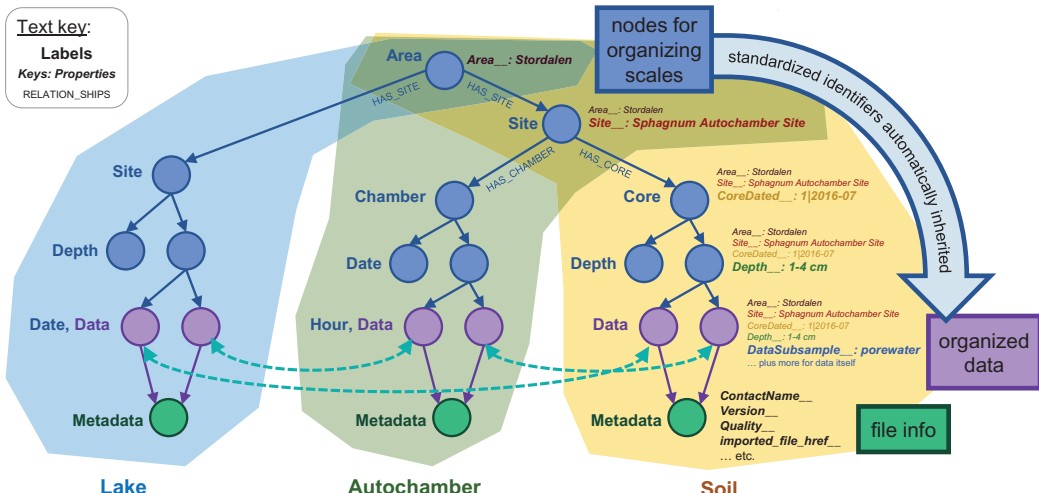

**Figure 3 Basic node structure of the IsoGenieDB.** This structure is shown for three representative dataset types: lake temperature timeseries (blue), autochamber (green), and soil core data (yellow). Nodes used purely for spatial and temporal organization are in blue, data-containing nodes are purple, and nodes containing file metadata are in green. Relationship labels (shown for the first few relationships in the tree) follow a similar naming scheme throughout the database. Data are stored as key:value properties within the nodes (shown for the soil data), with special standardized properties (denoted with keys ending in "__") storing sampling metadata related to each node's spatial-temporal placement. These standardized properties are automatically inherited from "upstream" in the scale-organized tree, while file Metadata nodes include their own standardized properties. This entire structure can be queried simultaneously to find derived relationships among data from different branches of the database (illustrated with curved dotted teal arrows).

location, date of sampling or field measurement, and general dataset type (e.g., gas fluxes, biogeochemical soil properties, or temperature timeseries), with the order of classification determined by the inherent nature of each data collection scheme (e.g., while soil cores and field sensors both exist at point locations, a soil core is collected on a particular day and can have data in several datasets across disciplines, whereas an installed field sensor typically only collects one dataset type over a span of many days). At the "root" of this structure is an Area node that represents the whole of Stordalen Mire. This Area node then branches to nodes representing increasingly specific spatial and/or temporal resolution, with highly-resolved Data nodes (representing small-scale spatiotemporally-separated data entities) finally appearing at the finest scales (Fig. 3). This data organization paradigm mimics the way in which sampling schemes are conceptualized within IsoGenie and similar projects, with sampling entities (including field instruments such as temperature sensors, as well as physical samples such as cores) located within named sites, each with their own sampling intervals that can then be matched across dataset types based on the date of sampling. In contrast to the default grouping of data within separate datasets, this location- and time-based data model provides a straightforward way to integrate spatiotemporally related data from different disciplines, aiding in the exploration of emergent phenomena. While this base structure is tree-like, other conceptual patterns also exist and are traced with other relationship structures. To track dataset origin and file metadata, each Data node is

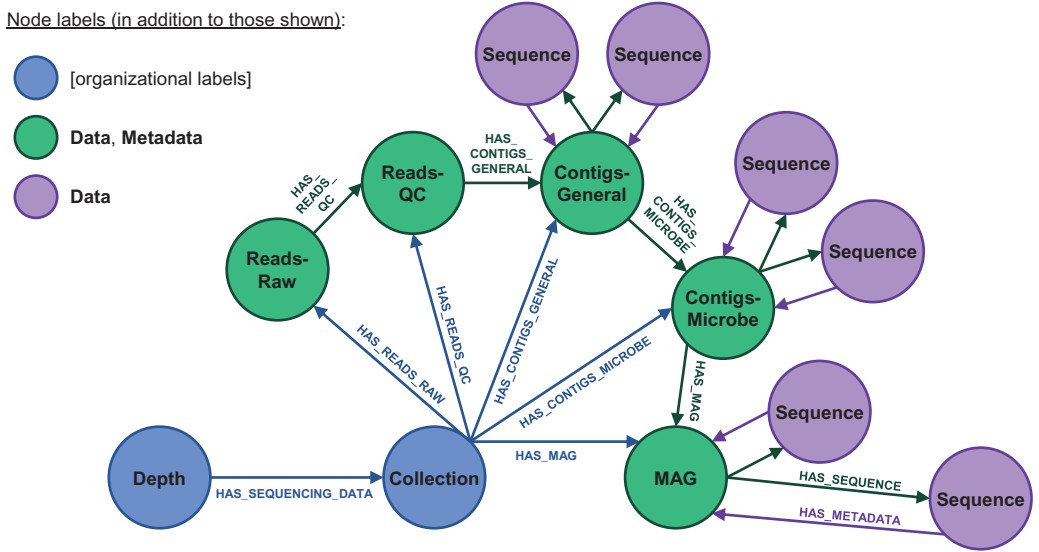

**Figure 4 Database node structure for microbial sequencing data.** These datasets include very large, detailed nucleic and amino acid sequences at varying quality levels. Raw reads are quality-checked (QC'd) and then binned into contigs, which are then used to derive fundamental knowledge on microbial ecology via metagenome-assembled genomes (MAGs). The contigs and MAGs, in turn, can be separated into individual sequences (purple). The IsoGenieDB stores these variously processed datasets and data entities in a node/relationship network that maps to the data processing pipeline, with very large sequence files stored as links to flat files in combined Data/Metadata nodes (green). These combined nodes also have HAS_METADATA relationships originating from the Sequence nodes, consistent with the Data→Metadata node structure used for other types of datasets (Fig. 3), in addition to HAS_SEQUENCE relationships pointing in the opposite direction to mirror the data processing order. "Collection" nodes are also used to group data originating from the same sample (blue), acting as an additional organizational paradigm for cases where not all data processing steps are available to build the full pipeline node structure.

linked to a Metadata node, which includes source file metadata such as file paths, authors, version, and data quality (Fig. 3). More complex data processing pipelines, such as those used for –omics and spectral data processing, are described with networks of converging and diverging Data and Metadata nodes (Fig. 4) (*Barkstrom, 2010*).

# RESULTS

## Overview

The IsoGenieDB, which has >400,000 nodes, combines the IsoGenie Project's data into an integrated structure that serves as a centralized resource for the project. This structure can be used to cross-reference different disciplinary datasets, providing a way to quickly retrieve custom subsets of data to address specific research questions. The IsoGenieDB also includes a public-facing web portal (https://isogenie-db.asc.ohio-state.edu/) that allows users to browse datasets using several methods, including a Queries page, a Downloads page, and a Map Interface. The IsoGenieDB is also being expanded to cover the NASA-funded A2A project via the Archaea to Atmosphere Database (A2A-DB), which overlaps with the IsoGenieDB (Fig. 5) and is accessed through a separate web portal (https://a2a-db.asc.ohio-state.edu/; Map Interface shown in Fig. 2A).

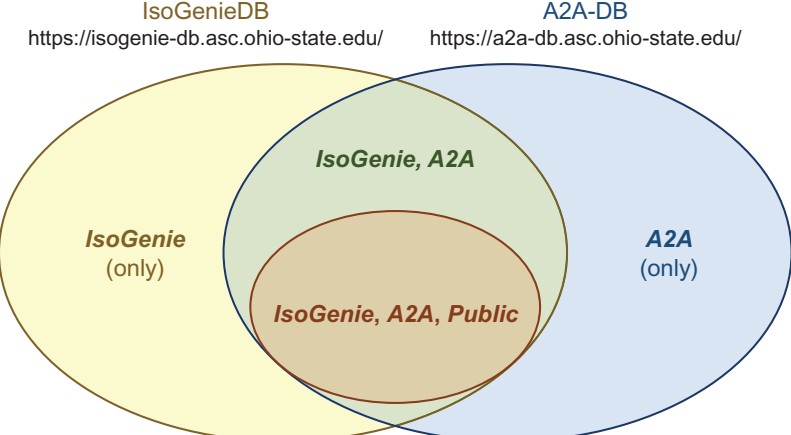

**Unified Graph Database**

**Figure 5 Integrated structure of the IsoGenieDB and A2A-DB.** A single unified graph database (overall Venn diagram) encompasses both the IsoGenieDB (yellow ellipse) and A2A-DB (blue ellipse). These databases share some of the same data (green region), a subset of which can be accessed by the general public (orange ellipse). Separate web portals (URLs under database names) provide the projects with access to their respective subsets of data, with public data access granted by the public-facing portions of each website. The access permissions for any given node are controlled by labels (bold italic text), which are used to filter the results of queries as part of the back-end of each website's Querying page (e.g., all queries from the A2A-DB web portal include a built-in filter that only returns those nodes with the label "A2A", and queries from the public web portals only return nodes with the label "Public").

## Data exploration using the IsoGenieDB and its web platform
### Interdisciplinary data queries

One strength of the IsoGenieDB is its ability to cross-reference data between seemingly disparate datasets. Specifically, the integrated graph structure allows cross-disciplinary queries (such as relating atmospheric conditions to microbial composition within a single query), enabling detailed exploration of data from multiple datasets simultaneously. This contrasts with most existing data repositories, which tend to keep data in separate datasets and/or be geared towards a particular discipline, creating a need to manually filter files based on sampling metadata and then manually recombine the data to find such relationships. Also, by enabling direct traversal of the relationship structure, the graph database architecture allows complex searches to be performed several orders of magnitude faster than in relational databases (*Neo4j, 2020*). This helps researchers to more efficiently find and compare data from different disciplines, enabling the discovery of new phenomena. Queries to the database can be either *simple* or *advanced*.

*Simple* queries are those that select data based on labels. These queries can be performed via the web portal's Querying page, which provides both live querying (in development) and the results of saved or "cached" queries. For example, filtering based on the Core label (Fig. S1) returns a table of all core nodes in the database, which can then be filtered based on the value of any column to allow the user to quickly retrieve core metadata. Querying based on multiple labels (available via live queries) provides another means of

filtering the results (e.g., selection of both Core and Biogeochemistry labels returns cores that have biogeochemistry data).

*Advanced* queries are those that require the input of a custom Cypher command via Neo4j's built-in browser (Fig. S2), allowing the user to filter data based on any custom criteria and then view the results as they appear within the graph structure itself. These might include, for example, identifying depths with pH < 5 that have microbial abundance data (this query would traverse the node structures for both the—omics data (Fig. 4) and soil porewater data (Fig. 3) that link to the same Depth nodes), or identifying multiple sites where specific microbial populations (as denoted with specific text) are present simultaneously over a specific time frame (which would involve searching for time-specific standardized sampling metadata among the Sequence nodes (Fig. 4)); further examples are provided in the "Discussion". The IsoGenieDB currently limits this advanced querying functionality to the database team (as the current command-line interface does not provide read-only access, nor does it restrict users from accessing private data from projects of which they are not members—for simple queries, the latter is accomplished on the back-end by filtering the query results to include only those nodes that are labeled with the project name [for the private portals] or as "Public" (for the public portals), as described in Fig. 5). Future development will focus on ways to remove this restriction.

### Downloading data

Publicly available data can be downloaded in their original format via the Data Downloads page, where datasets are listed as "packages" and made available for download (Fig. S3). These packages include the names and contact information for the data contributors, as well as links to any associated publications and README files describing the dataset. For spreadsheets, these packages also include CSV versions that have been formatted and had their metadata nomenclature standardized by the database team for import into the IsoGenieDB. This not only facilitates transparency of the data import process, but also provides easily script-readable versions of these files that other project members can quickly import into R, Matlab, or other analysis tools. Direct download of these raw, intermediate, and processed files is provided by the local web server, eliminating the need to query the database for retrieval.

Each download package also includes information on any updates or corrections to each dataset. As datasets are updated, they are assigned version numbers (formatted *x.y.z*) that separately track major ($x$) and minor ($y$) revisions by the original data generator, as well as versions of format-standardization performed by the database team ($z$). Previous versions remain downloadable as zipped archives. Separate from version numbers, each dataset is also assigned a Quality Level (Table 2), which serves as an indicator of the degree to which the dataset has been subject to error checking, blank corrections, calibrations, peer review, and other quality control.

### Browsing sampling sites

Sampling locations can be viewed through the Map Interface (Fig. 2), which displays information from GPS-annotated Site and Core nodes as points on an interactive map.

**Table 2 Data quality levels, used to denote quality control (QC) and processing status.**

| Quality level | Description | Example (using geochemical spectral data, such as FTIR) |
|---|---|---|
| 0 | Raw data | Raw, unprocessed spectra directly from the instrument |
| 1 | Data with basic QC. Internally consistent with similar datasets from the same lab | Spectra with baseline corrections, and chemical abundances derived therefrom |
| 2 | Data with extensive, externally-checked QC. In addition to undergoing the QC necessary for Quality 1, the data has also been reviewed for external consistency with similar data from other labs | Spectra with baseline corrections and derived chemical abundances, which have passed peer review for publication in a manuscript |

Each point can be clicked to display a popup, which lists the available types of datasets associated with that site or core (e.g., terrestrial geochemistry, autochamber measurements, etc.) along with photos of cores that can be downloaded. The displayed points can be filtered by habitat type classification, allowing selection of sites that belong to specific habitats. These points are overlain on an ultra-high-resolution drone image of a portion of the Mire (*Palace et al., 2018*).

### General information and outreach

Beyond data integration and discovery, the IsoGenieDB website also enables outreach by showcasing the research of the IsoGenie Project members through several educational pages (Fig. S4). These include a repository of >2,000 photos, which can be filtered by tags including general category (e.g., Core, People, Site, Landscape, etc.), habitat type (e.g., Palsa, Bog, Fen), site name, sampling year, and others. Additional pages include frequently asked questions relating to the database design, a list of project-related publications, and an interactive graphic of the IsoGenie Project members.

## DISCUSSION

### Advantages of the graph structure

Unlike simple file repositories, the IsoGenieDB graph structure integrates disparate datasets into a single framework, while offering more flexible expansion and faster querying than relational databases. This allows one to use a single query to filter data from various sources based on custom criteria and extract only the data that matches that filter. In contrast, accomplishing the same task in existing environmental repositories (such as LTER or DataONE) often necessitates downloading multiple source files and then manually filtering them, a tedious process. Even if some built-in filtering is available, this capability is often limited in other databases (e.g., in JGI's IMG, although datasets can be filtered by habitat, they cannot be filtered by specific field site, and even the habitat labels are often unavailable or described incorrectly). Being able to filter data based on *any* of several criteria with just a few clicks (for simple queries) or a few lines of code (for advanced queries) enables more efficient data discovery, allowing investigators to explore new hypotheses on the fly.

The structure of the IsoGenieDB is also ideal for straightforwardly storing data at multiple processing stages. Many datasets include raw, processed, and summarized data,

with the latter particularly helpful for interdisciplinary investigation, yet this multiplicity poses challenges for interdisciplinary data exploration. The IsoGenieDB addresses this problem by storing data in a framework that mirrors the data processing steps (Fig. 4). Because each data entity's origin is explicitly reflected in the graph structure, database users can get a general idea of each data entity's origin and level of processing even for data from outside their main discipline. This contrasts with discipline-specific databases such as those in NCBI, which are built for bioinformaticians and can therefore pose a challenge for other scientists who are looking for specific types of data products (e.g., raw vs. processed reads, or 16S data vs. metagenomes) for collaborative meta-analysis, but who may be unfamiliar with the names of the methods (e.g., pyrosequencing) used to generate these different data products.

Three querying examples below make use of these unique features:

1. (simple query) Retrieve metadata for all soil cores collected over the course of the project, regardless of which lab recorded the collection of the core (Fig. S1). This is helpful for planning future field sampling priorities based on which sites and habitats have been over- or under-sampled in the past, without the need to manually consolidate this information from multiple field sheets. Although the ISA framework (https://www.isacommons.org/) (*Sansone et al., 2012*) specializes in tracking this type of metadata, it stores the metadata within tabular or JSON files, and thus does not make use of the fast querying capabilities of graph databases.

2. (advanced query) Retrieve biogeochemistry data from collapsed palsa sites sampled on dates when the maximum air temperature recorded by ANS was >25 °C (Fig. S2). When combined with similar queries that filter based on other temperature ranges, this could provide insight on possible temperature-dependent biogeochemical processes warranting further investigation. The data necessary for answering this type of question might be findable within a large environmental repository such as DataONE (https://www.dataone.org/); however, this data would most likely be in separate files, which would then need to be individually filtered to find the dates matching the temperature criteria and then to find the biogeochemistry data from samples collected on these dates. In contrast, the IsoGenieDB can quickly retrieve these matched data with a single query.

3. (advanced query) Find all samples where a specific microbe was found, based on the presence or absence of relationships connecting metagenomic data to a specific MAG, and then compare the biogeochemistry of these samples to those where that microbe was *not* found. This query makes direct use of the node/relationship structures corresponding to data processing steps, specifically those connecting physical samples to metagenomes to MAGs (Fig. 4). To do this "traditionally", one would need to utilize BLAST to compare the specific MAG(s) to all of the samples, parse the results to identify non-matching sample(s), and then retrieve the sample metadata. Finally, these samples and their metadata outputs would need to have their biochemistry records retrieved. While relatively straightforward, this processing requires BLAST, a bioinformatic skillset to parse results, and numerous files stored in disparate formats (e.g., non-sample metadata, biochemistry records, sequencing files).

## Balancing efficiency, consistency, and flexibility of data import

One of the challenges of building the IsoGenieDB has been data heterogeneity, which makes it difficult to efficiently import data at a granular level without making it too "messy" for the end user. More specifically, a balance is required between importing all data as-is (at the cost of reduced database searchability), or conversely, thoroughly standardizing all of the data (at the cost of more time-consuming data import and potential for human error). To balance these conflicting needs, the IsoGenieDB import code prioritizes standardizing metadata essential to determining each node's placement (e.g., site names, date formats, etc.). These essential metadata properties, denoted with double underscores (e.g., "Site__"), are automatically propagated down the data classification tree (e.g., a node for a core section is automatically labeled with the core name and sampling date). This ensures that each node is given consistent labeling and placement regardless of dataset origin, allowing for greater ease and flexibility of querying the database. Meanwhile, more detailed measurement types and units remain in the original format contributed by each project member. Because the labels identifying node and relationship types are standardized across the database, data from different sources can be simultaneously queried and then exported as a unified data table, where the specific variables of interest can then be easily converted into common units for interdisciplinary data exploration (e.g., for principal components analysis or modeling inputs).

The use of standardized spatial and temporal information for node placement, combined with keeping the rest of the data in its original format, also allows the codebase for the database to be easily reused for new datasets with only minimal metadata standardization of the contributed data files. In a mid-size project such as IsoGenie (~$1 M/yr divided across 13 laboratories), ensuring a low upkeep burden is an essential feature of the IsoGenieDB.

## Future developments

### Improved cross-linkages between the graph database, web platform, and established data standards

Currently, the graph database and web portals share information (e.g., cached query results are regularly output to the Queries page, the Map Interface derives the map points and their associated properties from the sample metadata stored in graph nodes, and file metadata stored on the Data Downloads page matches that stored in the Metadata nodes of the graph database), but these components are not dynamically connected. Future improvements to the server architecture will enable the web server to communicate with the graph database in real time, allowing these pages to remain in sync with the graph database and to be automatically updated as new data are added.

As part of this automation, the metadata for individual datasets, listed in the Data Downloads page of the web portal, will be made to better conform with FAIR data standards. This will include functions to assign each dataset a DOI, to generate an EML file for each dataset (*Jones et al., 2019*) based on the file metadata stored in the graph database, and to create properties that store sample metadata using more broadly-used
ontologies, such as ENVO (*Buttigieg et al., 2013*, *2016*), so that the data can be more readily usable for global analyses.

### Enhanced user accounts providing secure access to advanced querying

Future improvements to the web portal will also make advanced querying available to all users. This will require the deployment of two major security upgrades, which will be implemented via new module(s) added to the back-end of the web portals:

1. Ensure that all user-submitted queries are read-only. This will involve three separate safeguards to greatly reduce the chances of unauthorized writes to the database: (a) refuse all Cypher statements that contain "write" keywords such as CREATE, DELETE, MERGE, and SET; (b) route the query to Neo4j through a read-only connection; and (c) automatically back up the database at regular intervals so that it can be restored if both of the previous safeguards fail (these regular backups are already in place).

2. Ensure that access to private data remains restricted to the project that generated it. This is specific to the case where a single database houses data from more than one project, as is the case for IsoGenie and A2A project collaboration, and would be relevant to other "nested" project sets where one project builds on the first and includes additional datasets that are not explicitly shareable with the personnel of the nested project. Project-specific access is currently being accomplished for simple queries by including the labels "IsoGenie", "A2A", and/or "Public" in the search criteria for the pre-generated Cypher MATCH queries requested from the private IsoGenie, private A2A, and public web portals, respectively (these labels are added during data import; Fig. 5). Controlling data access for advanced queries would make use of these same labels, and might be accomplished in the following ways: (a) create entirely separate graph databases for each user group, which would be regularly updated, based on project-access node labels in the current shared database; (b) use string replacement to add these labels to all MATCH clauses submitted by users; or (c) develop an "intermediate query" interface, which would keep the same back-end access label filtering used for simple queries, while allowing users to choose among a greatly expanded set of querying options.

### Storing experimental results as graph structure

The graph structure of the IsoGenieDB not only more fully allows researchers to address their own discipline-specific questions, but also can facilitate the identification of emergent relationships in the data. Currently, all data in the IsoGenieDB are stored as properties within nodes that are organized based on sampling and experimental schemes. While this makes possible the simultaneous querying and export of multiply sourced data, other tools still must be used to explore ecological relationships within this data. A future goal of the IsoGenieDB is therefore to explicitly encode experimental results in the broader node structure itself (e.g., to add relationships connecting plant species to sites, or

microbes to the chemical compounds they metabolize), allowing the direct use of the graph structure for network-based data mining (*David et al., 2015*; *Yoon, Kim & Kim, 2017*).

### Sharing across projects

Because the graph database framework allows easy addition of new sites and dataset types, this allows the database to be expanded to cover multiple projects with overlapping datasets. This functionality is currently being implemented with the A2A-DB, which includes a subset of the IsoGenie Project's data, along with data from the A2A project's other study sites (each of which has its own Area node). The IsoGenieDB and A2A-DB share the same back-end graph database, in which separate access for each project and the general public is controlled with dedicated node labels (IsoGenie, A2A, and Public) (Fig. 5). This system offers the advantage that updates to shared data from one project are automatically accessible by other project(s) that have access to the same data, while separately coded web interfaces (with separate sets of authorized users) prevent private datasets from being shared outside their respective projects.

Another goal of the IsoGenieDB is for its import code to be more easily reusable by other projects, such that other groups can easily generate their own databases based on the IsoGenieDB's basic design. The codebase already separates the scripts used for dataset-specific header standardization from those used for import to the database, allowing the import scripts to be reusable for different datasets of similar structure with minimal modification. Future development will convert these import scripts into functions that can accept custom parameters for specific node labels and other high-level metadata, thus removing the need for these small modifications to the scripts. Over the longer term, these functions may be further consolidated into a single function that can create node trees (Fig. 3) with any arbitrary number of "levels," which would be read from structured "node map" objects (one of which would be associated with each dataset). By making it easier for other projects to adapt the IsoGenieDB import code to suit their own data, these developments will improve the ability of the IsoGenieDB to serve as a model for other ecosystems databases.

## CONCLUSIONS

The IsoGenieDB addresses the challenges of interdisciplinary data integration by organizing large, interdisciplinary project data into a novel platform that can incorporate nearly any type of dataset, and that allows scientists to explore data through the inherent conceptual relationships between data entities both within and across datasets. While the graph structure allows for the flexible integration of heterogenous datasets, the use of standardized labels and relationship types ensures that the metadata describing each dataset remains relatively homogenous throughout the database, allowing detailed querying across all these multi-scale elements. Querying the data for specific features of interest that were not associated during initial collection or primary analyses allows discovery of new relationships. By storing all IsoGenie Project data in this integrated framework, and being able to augment with data from various other sources (such as the 100-year datasets from ANS (*Callaghan et al., 2010*; *Jonasson et al., 2012*) and paleoclimate

records (*Kokfelt et al., 2009*, *2010*)), the IsoGenieDB is capable of describing all elements of an ecosystem—from nanoscale to planetary, from microseconds to centuries, and from microbes and viruses to plants and gases. This structure also lends itself to the automated generation of metadata in more broadly-used nomenclature and format standards (as is being planned for future improvements), which will allow users to interact more dynamically with the database and facilitate FAIRer data sharing via the web portal. Because of these features, the IsoGenieDB can serve as an example ecosystems database for managing and storing nearly any type and amount of data.

## ACKNOWLEDGEMENTS

We gratefully acknowledge a decade of support from the Abisko Naturvetenskapliga Station (Abisko Scientific Research Station, Sweden) for annual field campaigns and on-site experiments, and Station member Annika Kristoffersson for contribution of Station climate data to the IsoGenieDB. We also thank Justin Slauson at the Ohio State University College of Arts and Sciences Technology Services (ASC-Tech) for assisting with initial establishment and ongoing technical support for the IsoGenieDB and A2A-DB web server.

The *IsoGenie Project Coordinators* are: Scott Saleska (University of Arizona), Virginia Rich (The Ohio State University), Patrick Crill (Stockholm University), Jeff Chanton (Florida State University), Gene Tyson (University of Queensland), Ruth Varner (University of New Hampshire), Matthew Sullivan (The Ohio State University), Steve Frolking (University of New Hampshire), Changsheng Li (University of New Hampshire), Eoin Brodie (Lawrence Berkeley National Lab), and William Riley (Lawrence Berkeley National Lab).

The *IsoGenie Project Team* includes the Coordinators and the following members:

- *Field Teams:* Darya Anderson (University of Arizona), Hanna Axén (Abisko Scientific Research Station), Kathryn Bennett (University of New Hampshire), Jeff Chanton (Florida State University), Patrick Crill (Stockholm University), Sky Dominguez (University of Arizona), Jessica Ernakovich (University of New Hampshire), Florencia Fahnestock (University of New Hampshire), Steve Frolking (University of New Hampshire), Anthony J. Garnello (current location: Northern Arizona University), Suzanne Hodgkins (The Ohio State University), Moira Hough (University of Arizona), Joachim Jansen (current location: Stockholm University), Robert Jones (current location: Cold Regions Research and Engineering Laboratory), Eun-Hae Kim (current location: Ventana Medical Systems), Changsheng Li (University of New Hampshire), Tyler Logan (current location: Numbers 4 Nonprofits), Tamara Marcus (University of New Hampshire), Samantha McCabe (The Ohio State University), Carmody McCalley (current location: Rochester Institute of Technology), Amelia McClure (current location: Horsley Witten Group), Rhiannon Mondav (current location: Uppsala University), Apryl Perry (University of New Hampshire), Nicole Raab (current location: Battelle), Virginia Rich (The Ohio State University), Scott Saleska (University of Arizona), Kristina Solheim (current location: Viosimo, LLC), Peter Tansey (University of New

Hampshire), Gareth Trubl (current location: Lawrence Livermore National Laboratory), Gene Tyson (University of Queensland), Ruth Varner (University of New Hampshire), Martin Wik (Stockholm University), and Rachel Wilson (Florida State University).

- *Laboratory & Analytic Teams:* Darya Anderson (University of Arizona), Ben Bolduc (The Ohio State University), Joel Boyd (University of Queensland), Eoin Brodie (Lawrence Berkeley National Lab), Sophia Burke (University of New Hampshire), Kuang-Yu Chang (Lawrence Berkeley National Lab), Jeff Chanton (Florida State University), William Cooper (Florida State University), Alex Cory (Florida State University), Patrick Crill (Stockholm University), Dylan Cronin (The Ohio State University), Kelsey Crossen (current location: Tehachapi Unified School District), Jia Deng (University of New Hampshire), Sky Dominguez (University of Arizona), Ellen Dorrepaal (Umeå University), Joanne Emerson (current location: University of California, Davis), Paul Evans (University of Queensland), Florencia Fahnestock (University of New Hampshire), Steve Frolking (University of New Hampshire), Anthony J. Garnello (current location: Northern Arizona University), Suzanne Hodgkins (The Ohio State University), Robert Hoelzle (University of Queensland), Moira Hough (University of Arizona), Beth Huettel (Florida State University), Bonnie Hurwitz (University of Arizona), Robert Jones (current location: Cold Regions Research and Engineering Laboratory), Ulas Karaoz (Lawrence Berkeley National Lab), Eun-Hae Kim (current location: Ventana Medical Systems), Josh Kolengowski (The Ohio State University), Changsheng Li (University of New Hampshire), Fen Li (The Ohio State University), Tamara Marcus (University of New Hampshire), Miguel Martinez (current location: Ensenada Center for Scientific Research and Higher Education), Samantha McCabe (The Ohio State University), Carmody McCalley (current location: Rochester Institute of Technology), Amelia McClure (current location: Horsley Witten Group), Rhiannon Mondav (current location: Uppsala University), Isabel Morales (The Ohio State University), Duc Nguyen (current location: Linköping University), Clarice Perryman (University of New Hampshire), Nicole Raab (current location: Battelle), Virginia Rich (The Ohio State University), William Riley (Lawrence Berkeley National Lab), Scott Saleska (University of Arizona), Caitlin Singleton (University of Queensland), Matthew Sullivan (The Ohio State University), Malak Tfaily (University of Arizona), Gareth Trubl (current location: Lawrence Livermore National Laboratory), Gene Tyson (University of Queensland), Ruth Varner (University of New Hampshire), Nathan VerBerkmoes (current location: STEM Educational Consultations, Inc.), S. Rose Vining (current location: State of Alaska Department of Environmental Conservation), Rick Wehr (University of Arizona), Martin Wik (Stockholm University), Rachel Wilson (Florida State University), Katelyn Winters (The Ohio State University), Ben Woodcroft (University of Queensland), Gregory Zane (current location: University of Washington), and Ahmed Zayed (The Ohio State University).

The Archaea to Atmosphere (A2A) Project Team includes the following members: Ben Bolduc (The Ohio State University), Rob Braswell (Applied Geosolutions),

Sophia Burke (University of New Hampshire), Eleanor Campbell (University of New Hampshire), Patrick Crill (Stockholm University), Jessica DelGreco (University of New Hampshire), Jia Deng (University of New Hampshire), Florencia Fahnestock (University of New Hampshire), Justin Fisk (Applied Geosolutions), Christina Herrick (University of New Hampshire), Suzanne Hodgkins (The Ohio State University), L. Jamie Lamit (Syracuse University), Carmody McCalley (Rochester Institute of Technology), Michael Palace (University of New Hampshire), Apryl Perry (University of New Hampshire), Clarice Perryman (University of New Hampshire), Virginia Rich (The Ohio State University), Joanne Shorter (Aerodyne Research), Franklin Sullivan (University of New Hampshire), Nathan Torbick (Applied Geosolutions), Ruth Varner (University of New Hampshire), and Beth Ziniti (Applied Geosolutions).

### Funding

This study was funded by the Genomic Science Program of the United States Department of Energy Office of Biological and Environmental Research, grants DE-SC0004632 and DE-SC0010580; the NASA Interdisciplinary Research in Earth Science (IDS) program, grant # NNX17AK10G; Vetenskaprådet (Swedish Research Council, VR) to Patrick M. Crill (2007-4547 and 2013-5562); and a National Science Foundation "iVirus" grant (ABI #1759874) and Gordon and Betty Moore Foundation Investigator Award (#3790) to Matthew B. Sullivan. The funders had no role in study design, data collection and analysis, decision to publish, or preparation of the manuscript.

### Grant Disclosures

The following grant information was disclosed by the authors:
United States Department of Energy Office of Biological and Environmental Research: DE-SC0004632 and DE-SC0010580.
NASA Interdisciplinary Research in Earth Science (IDS): NNX17AK10G.
Vetenskaprådet (Swedish Research Council, VR): 2007-4547 and 2013-5562.
National Science Foundation "iVirus": ABI #1759874.
Gordon and Betty Moore Foundation Investigator Award: #3790.

### Competing Interests

The authors declare that they have no competing interests.

### Author Contributions

- Benjamin Bolduc conceived and designed the experiments, performed the experiments, analyzed the data, prepared figures and/or tables, authored or reviewed drafts of the paper, and approved the final draft.
- Suzanne B. Hodgkins conceived and designed the experiments, performed the experiments, analyzed the data, prepared figures and/or tables, authored or reviewed drafts of the paper, and approved the final draft.

- Ruth K. Varner conceived and designed the experiments, analyzed the data, authored or reviewed drafts of the paper, obtained funding, and approved the final draft.
- Patrick M. Crill analyzed the data, authored or reviewed drafts of the paper, obtained funding, and approved the final draft.
- Carmody K. McCalley analyzed the data, authored or reviewed drafts of the paper, and approved the final draft.
- Jeffrey P. Chanton analyzed the data, authored or reviewed drafts of the paper, obtained funding, and approved the final draft.
- Gene W. Tyson analyzed the data, authored or reviewed drafts of the paper, obtained funding, and approved the final draft.
- William J. Riley analyzed the data, authored or reviewed drafts of the paper, obtained funding, and approved the final draft.
- Michael Palace analyzed the data, authored or reviewed drafts of the paper, and approved the final draft.
- Melissa B. Duhaime analyzed the data, authored or reviewed drafts of the paper, and approved the final draft.
- Moira A. Hough analyzed the data, authored or reviewed drafts of the paper, obtained funding, and approved the final draft.
- Scott R. Saleska conceived and designed the experiments, analyzed the data, authored or reviewed drafts of the paper, obtained funding, and approved the final draft.
- Matthew B. Sullivan analyzed the data, authored or reviewed drafts of the paper, obtained funding, and approved the final draft.
- Virginia I. Rich conceived and designed the experiments, analyzed the data, prepared figures and/or tables, authored or reviewed drafts of the paper, obtained funding, and approved the final draft.

## Data Availability

The code used to interact with the IsoGenieDB, including standardizing and importing data and performing queries, is available at BitBucket https://bitbucket.org/MAVERICLab/isogeniedb-tools/.

The IsoGenie Database, and all public data deposited therein, is available at https://isogenie-db.asc.ohio-state.edu/.

The A2A Database and associated public data is available at https://a2a-db.asc.ohio-state.edu/.

## Supplemental Information

Supplemental information for this article can be found online at http://dx.doi.org/10.7717/peerj.9467#supplemental-information.

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
