# Peer review of "The IsoGenie database: an interdisciplinary data management solution for ecosystems biology and environmental research"

_PeerJ, doi:10.7717/peerj.9467_

## Round 0.1 · original submission · Major Revisions

The manuscript is well written. However, it would be improved by extending the existing examples to describe how certain decisions were made and why potential users should consider IsoGenie over other options. Clarify the terminology in the introduction and be inclusive of other fields, when appropriate, to highlight the interdisciplinary appeal of the database. I look forward to seeing the revisions.

Reviewer 1 ·

Basic reporting

The paper is relatively clearly written. The literature is well referenced and the figures are relevant, high quality and well described.

The paper describes an online data management solution for environmental (!) and ecosystem ecology research together with a web platform for the IsoGenie project consortium with 13 laboratories.

The title is somehow misleading and should include the word “environmental”, because the majority of datasets listed in Table 1 do not refer to biological data. Proposal: “…. for ecosystems biology and environmental research”.

The introduction broadly describes the landscape of data repositories, online data management platforms and bioinformatics processing tools in the field of environmental and life sciences. It is also explaining some of the challenges to process and integrate data across discipline boundaries.

Nevertheless the introduction should be generally improved and better distinguish between basic terms like “data”, “data processing”, “datasets” and their processing and define “data types” (see “myriad data types” line 168, but https://en.wikipedia.org/wiki/Data_type) and “IsoGenie data types” as well as “metadata” in the sense of this paper. This is a pre-requisite to make it more clear what is the innovative power of the approach to create a “data structure to mimic the physical relationships between the data” (line 138, better datasets?) and “mirror the physical relationships and workflow between datasets” (line 155).

The introduction should also refer to some of the existing technical as well as content schemas, standards and norms. These (de facto) standards/ schemas for data exchange do not only exist in microbiology as suggested in line 106 but in other fields in biology, ecology and geosciences (e.g. like ISO norms, INSPIRE, ABCD, EML, ISA Abstract Model, MIxS Scheme). They give a certain level of abstraction and provide frameworks with rules and guidelines to integrate data in common interdisciplinary platforms and make them better exchangeable and reusable (FAIR data) also outside of joint (large) research projects and their “data silos”. The introduction also does not refer on approaches in using ontologies like ENVO http://www.obofoundry.org/ontology/envo.html with structured parameters to lower the boundaries between heterogeneous datasets.

Chapter “Results”

The chapter describes the IsoGenie project with A2A project addition (datasets with microbial communities), the data queries on the IsoGenie graph database as well as the web platform. Especially the “interdisciplinary data queries” should confirm the strength of graph data bases. This paragraph (line 233 until 251) is central and should be extended. Perhaps the author could add a diagram as in Yoon et al. (2017, Fig. 1) indicating the performance of the IsoGenieDB?

The access permission organised according nodes and the security risks shortly mentioned in line 248 and in legend to Figure 3 is not clear to the reviewer.

Chapters “Discussion” and “Conclusions”

Line 307: “discipline-specific jargon”, please see remarks on standards and ontologies above
Line 307: I would not regard databases in NCBI as “specialist databases”.

Line 323 ff. This is a good example for a complex query. Is it realised by the advanced query described under line 242 ff.? If so please refer to figure 5 there.

The chapter “Conclusions” refer again to the (future) advantages of the Graph database solution with the data structure set up for the IsoGenie project. The planned extensions for the advanced queries and for administration of user accounts and rights management should be better described. Some hints for future IsoGenieDB guided publication of FAIR data files using e.g. functions for DOI assignment and broadly accepted schemas for (meta-) data exchange might be envisaged. The current data download page (well organised with versioned download packages and quality control!) and browsing sampling site option is at the moment held separately from the graph database. Are there plans to change this?

The “Acknowledgements” could be shortened by linking websites where the project coordinators and team members as well as lab and analytic teams are listed.

Experimental design

The paper describes the design of a new database application and the web platform where the database is integrated and the set up and use of the database. This is done as an original primary research approach in life sciences and environmental sciences and - as such - meets the scope of the journal.

The original design of the IsoGenieDB (based on graph structure; powered by Neo4j) is visualised with the basic nodes for organizing scales (Fig. 4) and – more special – with the node structure for microbial sequencing data with “collection node” (Fig. 5). This design seems appropriate and is a useful precision of the property graph model to organize heterogeneous datasets from environmental research at a given geographic area not only according their metadata but also following the “has” relationships between them, the area, sites, date as well as physical samples as cores and create thereby nodes for organizing scales and a stable data network with standardized identifiers automatically inherited.

The design of the database and the programming code including the documentation of the “IsoGenieDB import code” and “metadata nomenclature standardized by the data team” mentioned in line 257 should be deposited in an open source code repository like https://gitlab.com/gitlab-org to fulfil the FAIR principles for data and research software.

Validity of the findings

The implementation and description of such a Graph database for heterogeneous datasets from environmental and ecosystems ecology research is challenging and worth to be published.

The IsoGenieDB is a valid approach to organise and process datasets and data in an interdisciplinary project. Its proper use in the IsoGenie + A2A project as pilot is certainly of great advantage. At the moment, however, the query options realised with Cypher are very limited and the results seen on the public web site are scarce and might be not appropriate for further scientific analysis in a world of open data.

There are also online services to register software tools, services for the life sciences and data repositories, e.g., https://bio.tools/ and https://www.re3data.org/. The IsoGenieDB implementations, other IsoGenie tools as well as the data repository with the advanced data download options are worth to be registered there.

For the future the authors should explore how to add certain standard scheme elements and further naming conventions to the automatic labelling with core names, sampling data etc. Adding labels to the graph nodes compliant to core elements of existing community-accepted metadata standards might possibly facilitate further global analyses of the datasets, e.g. with online tools like https://www.gfbio.org/de/visualize in a world of open science.

The “Conclusions” should be extended (see under basic reporting).

Reviewer 2 ·

Basic reporting

This manuscript was well written and easy to understand; however some more explicit examples (e.g. of specific queries) of how the database could serve as a model for other cross-scale ecosystem studies would improve the manuscript by providing some more solid grounding. Although the database is clearly well designed from the description, the lack of these concrete examples makes it challenging to envision the advantage of this approach over other existing approaches.

Experimental design

Cross-scale ecosystem studies involving heterogenous data from multiple research groups are increasingly becoming the norm. Increasing attention is paid to the importance of data management and the ability to best utilize available data across studies and researchers. As described, this database could serve as a model for other similar projects.

Validity of the findings

While the design of this system is clearly well thought out and effective (especially given the research that has been supported and enabled by it), it would helpful for those considering using it as a model database if more detail was provided on how critical design decisions were made. For example, why was the decision made to use a hierarchical organization based on location and data type, followed by information on spatial and temporal resolution? This piece of information would help readers to easily identify whether this approach could be useful for their project and potentially to identify modifications that would need to be made to suit their research questions and vision.

---

## Round 0.2 · accepted · Accept

Please see minor edit from Reviewer 1 about Fig 3 being cited before Fig 2.

Reviewer 1 ·

Basic reporting

After the revision the paper is now very clear, well-structured and unambiguous. The reviewer is especially appreciating the revised introduction with the new terminology chapter and that the software developers are documenting pieces of code and make the code freely available under https://bitbucket.org/MAVERICLab/isogeniedb-tools/ .

Experimental design

The experimental design, results and the discussion, especially on the advantages of graph structures and the ideas on future developments are well done. Only a small comment: Figure 3 is cited before Figure 2.

Validity of the findings

The paper is of great interest for interdisciplinary teams in microbial and ecosystem sciences who are looking for a concept and realisation of a data management and exploration platform. The IsoGenieDB described in detail (especially the figures 3 and 4 are very informative) with its flexible architecture might be an useful model and example for processing and accessing heterogeneous ecosystems biology and environmental research data.